# Physicochemical Aspects and Sensory Profiles as Various Potential Factors for Comprehensive Quality Assessment of Nü-Er-Cha Produced from *Rhamnus heterophylla* Oliv.

**DOI:** 10.3390/molecules24183211

**Published:** 2019-09-04

**Authors:** Le Wang, Shusheng Fan, Xiaoping Wang, Xiuhuan Wang, Xin Yan, Dongjie Shan, Wuqing Xiao, Jiamu Ma, Yanran Wang, Xiao Li, Xiao Xu, Gaimei She

**Affiliations:** School of Chinese Materia Medica, Beijing University of Chinese Medicine, Fangshan District, Beijing 102488, China (L.W.) (S.F.) (X.W.) (X.W.) (X.Y.) (D.S.) (W.X.) (J.M.) (Y.W.) (X.L.) (X.X.)

**Keywords:** Nü-Er-Cha, physical parameters, flavonol glycosides, hydroxybenzoic acids, sensory evaluation, inner relationship, TOPSIS

## Abstract

Nü-Er-Cha, produced from the leaves of *Rhamnus heterophylla* Oliv., is known as an herbal tea and used in the treatment of bleeding, irregular menstruation and dysentery. A method was developed for the quality assessment of herbal tea, Nü-Er-Cha, adopting physical parameters, chemical constituents and sensory profiles as various potential factors. Their inner relationship was mined by multivariate statistical analysis tools, and the three factors were integrated by a technique for order preference by a similarity to ideal solution (TOPSIS) approach to comprehensively analyze the characters of Nü-Er-Cha. Viscosity was also introduced to the physical parameter determination besides conductivity, pH and color. Seven common peaks of eight batches of Nü-Er-Cha were marked by a high performance liquid chromatography (HPLC) fingerprint. They were further identified by HPLC mass spectrometry/mass spectrometry (HPLC-MS/MS) as hydroxybenzoic acids and flavanol glycosides. Fifty trained members participated in the sensory evaluation. Significant correlations between total sensory scores and conductivity, viscosity as well as pH were observed, a relatively innovative result for the quality assessment of herbal teas. The common peaks, belonging to hydroxybenzoic acids and flavanol glycosides, were mainly related to the color of infusions and leaves. The result of the TOPSIS analysis showed that S3 and S4 ranked as the top two in the comprehensive quality assessment. This may be related to rhamnetin triglycoside with a galactose/glucose and two rhamnoses, which had a higher peak response in S3 and S4 than that in the other samples. The present study may contribute to a better understanding of the relationship regarding physical properties, chemical composition and sensory profiles, and it may supply ideas for the comprehensive quality assessment of the herbal tea Nü-Er-Cha.

## 1. Introduction

Herbal teas, prepared with the leaves, flowers, seeds, fruits, stems and roots of plant species other than the leaves of *Camellia sinensis* (L.) Kuntze (Theaceae), are consumed globally with long histories and cultural traditions [1,2]. In recent years, more and more research about herbal teas treating health problems including digestive disorders, colds, hepatic disorders and cardiovascular disease has been reported [3], resulting in the herbal teas’ increasing demand. The quality of tea and herbal teas have an important effect on consumer preference. Sensory evaluation is a common approach for the quality assessment of tea and herbal teas [4,5]. Though it is classical, the results are subjective and easily affected by diverse factors. To objectively assess the quality of tea and herbal teas, various technologies have been developed, including E-tongue and E-nose digitize sensory traits [6]. These emerging technologies are relatively expensive for large sample sizes. HPLC and GC fingerprints are the most common approaches for consistency evaluation. Illustrations of chemotypes and calculation of the content of sensory-related and bioactive components via HPLC and GC technologies are frequently used in the quality assessment of tea and herbal teas [7]. Sensory evaluation coupled with the determination of the content of free amino acids, caffeine, total polyphenols, catechins, and so on has been widely applied to assess the quality of tea and herbal teas [8,9]. There have also been many reports that some physical parameters have effects on the sensory quality of tea. For example, viscosity reflects the mobility strength of liquid and has an effect on perceived taste intensity [10]. The potential of hydrogen (pH) is a measure of the acidity or alkalinity of solution. The structure of compounds may be affected by pH, resulting in the alteration of solubility, which could further change the taste and color [11]. Color is a comprehensive indicator of color, aroma, taste and shape, one whose modification will actually reflect the variation of chemical composition in tea infusions [12]. Mineral ions significantly impact conductivity, which could complex with polyphenols and result in changes in taste and color [13]. Consequently, physical parameters could characterize the sensory traits of tea and herbal teas. Physical parameters have frequently been used for the quality assessment of tea, while their application in herbal teas has been sporadic and lacking. In the present study, physical parameters, sensory profiles and chemical constituents were regarded as potential factors for the quality assessment of Nü-Er-Cha. The technique for order preference by similarity to ideal solution (TOPSIS) approach is one of the useful multi-criteria decision-making techniques for tackling ranking problems [14]. It has been applied in the comprehensive quality evaluation of traditional Chinese medicine and process optimization of food [15,16]. In order to comprehensively evaluate the quality of Nü-Er-Cha, a method which simultaneously takes physical parameters, chemical constituents and sensory profiles into consideration was developed, and the quality of eight batches of samples was ranked by the TOPSIS approach.

Southwest of China are the ethnic minority areas. Herbal teas are drunk more frequently by native people for multiple healthcare uses. As a part of our systematical investigation of herbal tea plants [17,18], a detailed study of Nü-Er-Cha was carried out. Nü-Er-Cha is produced from the leaves of *Rhamnus heterophylla* Oliv. belonging to the *Rhamnus* genus. It is mainly distributed in mountains, hills and rocky areas of the Yunnan, Sichuan, Guizhou, Hubei, Shaanxi and Gansu provinces. Nü-Er-Cha has a long history of drinking and treating bleeding, irregular menstruation and dysentery by native people in the Sichuan province. The tender leaves are brewed with water, and the infusions, which are yellowish green, light bitter and without apparent aroma, are usually drunk as tea by native people. The attention on Nü-Er-Cha has been relatively lacking. Our team systematically investigated chemical constituents from Nü-Er-Cha. Ten compounds were isolated and identified. A quick analysis of chemical constituents was further conducted by HPLC-MS technologies, showing that the total 28 compounds mainly belonging to phenolic acids, anthraquinones and flavonoids. The HPLC fingerprint of eight batches of Nü-Er-Cha was established, while their anti-oxidant activity was determined by a 2,2-diphenyl-1-picrylhydrazyl (DPPH) free radical scavenging assay. Partial least squares regression (PLSR) and a grey relational analysis (GRA) were used to study the relationship between the HPLC fingerprint and free radical-scavenging activity of Nü-Er-Cha, indicating that protocatechuic acid, kaempferol, emodin-8-*O*-α-l-rhamnoside and emodin were significantly related to radical-scavenging activity [19]. In the present study, we tried to deem physical parameters, chemical constituents and sensory profiles as potential factors for the quality assessment of Nü-Er-Cha and integrate these factors by a TOPSIS approach for a comprehensive character analysis of Nü-Er-Cha. 

## 2. Results and Discussion

### 2.1. Analysis on Sensory Profiles, Physical Parameters and Chemical Constituents of Nü-Er-Cha 

#### 2.1.1. Sensory Profiles, a Potential Factor for the Comprehensive Quality Assessment of Nü-Er-Cha

Fifty panelists evaluated the appearance, infusion, infused leaves, taste and aroma of eight batches of Nü-Er-Cha. The higher total scores were recorded in S1 and S2. It was clearly that S3 and S4 had higher scores for infusion and infused leaves, but the scores for taste and aroma were lower compared to S1 and S2 (Figure 1A). In terms of the 12 indicators, S1 and S2 had higher scores in rope, evenness, sweetness and clean aroma (Figure 1B), while higher scores of brightness, leaf color, infusion color, transparency and texture were present in S3 and S4. It was obviously observed that almost all the lower scores were recorded in S5, S6, S7 and S8. Therefore, the samples may be divided into three groups based on the results of sensory evaluation: Group I contained S1 and S2 with the highest scores. Group II consisted of S3 and S4. Group III comprised the rest of four batches (S5, S6, S7 and S8) with the lowest scores. 

#### 2.1.2. Physical Parameters, Another Potential Factor for Comprehensive Quality Assessment of Nü-Er-Cha 

The viscosity, conductivity and pH values of the eight samples were different (Table 1). It was clear that the values of viscosity varied from 1.5296 to 1.6045, which may have been related to the differential content and type of chemicals in infusions. The lowest value of viscosity among the samples was that of S2, and the highest value was present in S8, corresponding with those results in lubrication scores. The conductivity of water was 868 μs/cm, which was lower than that of infusions. This phenomenon may be explained by the formation of some insoluble constituents involving metal ions. The range of Dcond. (the differential values between the conductivity of tap water and infusions) was 40–113 μs/cm. The highest and lowest differential values were recorded in S2 and S6, respectively, and—correspondingly—the clear difference of taste scores between S2 and S6 could also be observed. The pH values of Nü-Er-Cha infusions ranged from 7.18 (S1) to 7.49 (S8). Similarly, the taste scores of S1 and S8 were significantly different. All the pH values of infusions were lower than that of water (7.92), possibly due to the extraction of constituents such as organic acids and polyphenols.

The color of Nü-Er-Cha was evaluated using the CIELab system of chromatic coordinates. The color showed noticeable differences among infusions (Table 1). The infusions of S1 and S2 possessed the highest values of L* and the lowest values of b*, which indicated that the color of two infusions were the lightest and the least yellow compared to that of other infusions. The negative values in a* were only present in infusions of S3 and S4, showing a more intense greenness. It was worth noticing that the color was significantly different between the infusions and leaves of Nü-Er-Cha. The infusions had much larger values of L* and lower values of a* and b*, compared to that of leaves, indicating that the infusions were lighter, greener and yellower. The leaves of S4 were the only ones with a negative value in a*. Both the leaves of S3 and S4 had relatively higher values of L* and b*, which showed that they were lighter and yellower than the color of other samples. In both infusions and leaves, S3 and S4 were obviously distinguished from the other batches of Nü-Er-Cha. The results were matched with that of sensory evaluation, in which S3 and S4 had the highest scores of leaves and infusion color. These phenomena may be caused by the differences in kinds and contents of chemical composition present in infusions and leaves. Many compounds were in leaves, and most of the compounds in the infusions were water-soluble. For example, chlorophyll and carotenoids were the mainly color-related constituents in leaves, while the chemical constituents in infusions primarily consisted of flavonol and its glycosides [20].

#### 2.1.3. Chemical Constituents, the Potential Factor for Comprehensive Quality Assessment of Nü-Er-Cha 

Seven common peaks were found in the HPLC fingerprint (Figure 2A). Peak 6 was chosen as the reference peak because of its high content, high intensity and moderate retention time in the chromatograms. Then, the relative retention time and relative peak area were measured. The results of relative peak area of common peaks are shown in Table 2. This method was validated through investigations on precision, repeatability and stability. The Similarity Evaluation System for Chromatographic Fingerprint of Traditional Chinese Medicine (Version 2004A, National Committee of Pharmacopoeia, China) was applied to evaluate the similarity of eight batches of Nü-Er-Cha. A value of the correlation coefficients approaching 1.0 indicated a perfect similarity of the samples, while low correlation coefficients had low similarity (Table 3). All the correlation coefficients between the reference chromatogram and S1, S2, S5, S6, S7 and S8 were larger than 0.8, showing a higher similarity. The highest value of correlation coefficients was observed between S3 and S4 (0.988), which indicated significant similarity. Meanwhile, the correlation coefficients between S3, S4 and the reference chromatogram were 0.548 and 0.634, respectively. There was a peak marked by a red square in Figure 2A, which had a higher response in S3 and S4 than that in the other samples. It was identified as rhamnetin triglycoside with a galactose/glucose and two rhamnoses belonging to flavonoids, which may explain the obvious differences between S3, S4, and the other samples. The similar differences between S3, S4, and the other samples were also observed in the sensory evaluation and physical measurements. 

Applying the same chromatographic conditions with the HPLC fingerprint, the analysis of the peaks was performed by high performance liquid chromatography-diode array detector-electrospray ionization-mass spectrometry/mass spectrometry (HPLC-DAD-ESI-MS/MS) along with their retention time, detected mass, molecular formula and MS/MS fragment ions in negative ion mode. The compounds whose mass differential values lower than 5 ppm were recognized as the best possible match. The total ion current (TIC) chromatogram of Nü-Er-Cha is shown in Figure 2B. The common peaks were presumed as: Malic acid (X1), gallic acid (X2), protocatechuic acid (X3), salicylic acid (X4), quercetin triglycoside with a galactose/glucose and two rhamnoses (X5), kaempferol triglycoside with a galactose/glucose and two rhamnoses (X6), and rhanmocitrin triglycoside with a galactose/glucose and two rhamnoses (X7). Apart from these common peaks, other compounds were also speculated. The conjectured compounds can be classified into three groups: Flavonoids, anthraquinones and phenolic acids (Table 4). The flavonol glycosides were the main constituents in Nü-Er-Cha, which were also widely present in tea [21]. Moreover, anthraquinones and phenolic acids may be related to the taste and color of Nü-Er-Cha infusions. 

### 2.2. The Inner Relationship among Physical Parameters, Chemical Constituents and Sensory Profiles Based On Multiple Statistics Analysis Approaches

#### 2.2.1. The Analysis Based on Pearson Correlation Coefficients

Pearson correlation coefficients were used to describe the relationship between the physicochemical profiles and sensory evaluation of Nü-Er-Cha. A good correlation was observed for most of the analyzed parameters (Figure 3). Apparently, correlations were observed between the total scores of sensory evaluations and DpH, Dcond, and viscosity (r = 0.84, −0.725, and −0.75, *p* < 0.05, respectively). The scores of tastes were related to both pH and viscosity (r = 0.8741, −0.759, *p* < 0.05). Both infused leaves and infusions were positively connected to L*1 and b*1 but negatively correlated to a*2. Some reasons could be used to account for these results. The pH may affect the dissolution and structure of flavor substances. The kinds and concentration of ions in infusions may result in different conductivities. Some of the ions could have combined with phenol group in infusions—therefore, the flavor of infusions could have been altered. Viscosity could reflect the intensity of infusions’ fluidity and affect on mouth-feel. The lower the value is, the more intense the lubrication is. 

It was evident that common peaks of the HPLC fingerprint were mainly related to infused leaves and infusions (Figure 3). The common peaks of X5, X6 and X7 were positively correlated with infusion (r = 0.748, 0.777, and 0.738, *p* < 0.05, respectively). Both the common peaks X3 and X4 were negatively correlated with infused leaves (r = −0.741 and −0.800, *p* < 0.05, respectively). The leaf color was negatively related with common peak X4 and positively related with common peaks X5, X6, and X7. The common peaks X3, X4, X5, X6 and X7 were speculated as hydroxybenzoic acids and flavonol glycosides. These indicated that hydroxybenzoic acids and flavonol glycosides may be the contributors of color of Nü-Er-Cha. The perspective mentioned above agrees with previous literature related to the color of green tea affecting by flavonol glycosides [22]. All the results mentioned above indicate that the sensory attributes were closely connected with physicochemical profiles.

#### 2.2.2. The Analysis Based On PCA

Principal component analysis (PCA), an unsupervised method, was performed on full data to determine the location of samples in the sensory and physicochemical profiles map. The PCA bi-plot (Figure 4) explained 70.13% of the total variability of the data-set, with the horizontal axis (PC1) representing 41.88% and PC2 (the vertical axis) accounting for the remaining 28.25%. Along the horizontal axis (PC1), the high and low scores of sensory evaluations were separated, with the lower scores’ samples being located on the left side of PC1 and the higher scores’ samples being found on its right side. It was obvious that S1 and S2 were distributed at the positive value of PC1 and PC2, and S3 and S4 were located at the positive value of PC1 and the negative value of PC2. Additionally, S5, S6, S7 and S8 were distributed at the negative region of PC1. Overall, the eight samples were found in different locations of the PCA plot, thus reflecting the distinctive sensorial and flavor differences among the samples. The physicochemical parameters including L*1, b*1, X5, X6 and X7 were close to infusion color and leaf color, which verified the high correlation. Unlike the parameters mentioned above, the viscosity, Dcond., a*, X1, X3 and X4 were located on the opposite side of sensory indicators, indicating a negative connection with sensory characteristics. The correlation mentioned above agreed with those of the Pearson correlation coefficient analysis. As observed, it was possible to discriminate among the different samples. All the samples might be divided into three groups: Cluster I included S1 and S2, cluster II contained S3 and S4, and the other samples were clustered into the third group. The separation was in good accordance with that of similarity evaluation of the HPLC fingerprint and sensory evaluation. 

#### 2.2.3. The Analysis Based On PLSR

Partial least squares regression (PLSR) is usually used to find the inner relationship between independent variables (X) and dependent variables (Y) [23]. In the present study, both the results of the Pearson correlation coefficient analysis and the PCA verified the close correlation between sensory traits and physicochemical profiles. Hence, the X matrix was composed of the physicochemical parameters, and the Y vector was constructed with the total scores of sensory evaluations. For the analysis, only the predictors with a variable importance to projection (VIP) value larger than 1 were included (Figure 5). The DpH, Dcond., viscosity, b*1, X4 and X7 were selected to further fit, the Y vector was composed of the total scores of the sensory evaluation, and the X matrix was concluding DpH, Dcond., viscosity, b*1 and X7. The equation was as follows: Y = 10.2704 + 0.387876 DpH – 0.247651 Dcond. − 0.31862 viscosity + 0.0819157 b*1 – 0.136589 X4 + 0.113899 X7 (R^2^ = 0.9311). 

### 2.3. Comprehensive Evaluation of Nü-Er-Cha by TOPSIS Method

The quality of Nü-Er-Cha was significantly affected by both sensory characteristics and physicochemical properties. The TOPSIS approach was performed for comprehensively evaluating the quality of Nü-Er-Cha. Significant differences in rankings were observed, when only one of the chemical constituents, physical parameters and sensory traits was used as indicator in the TOPSIS method (Table 5). Hence, it was insufficient to evaluate the quality of Nü-Er-Cha by a single aspect. The three aspects consisted of chemical constituents, physical parameters and sensory traits, and these were simultaneously used in the TOPSIS approach. As seen in Table 6, the best alternative (0.67131) had the nearest distance to the positive ideal (0.28362) and also the farthest distance to the negative ideal (0.57927). The final rankings were as follows: S3 > S4 > S1 > S5 > S6 > S8 > S7 > S2. According to the rankings, S3 and S4 had a higher quality. Besides their good performance in both physical parameters and the sensory analysis, the chemical constituents, especially rhamnetin triglycoside, may have made important contributions to this result. Despite having higher scores of sensory evaluations, S2 still got the lowest ranking in the comprehensive assessment, which may be attributed to the relative lower values of some physicochemical parameters. Compared to the method of quality evaluation depending on single aspect, the TOPSIS approach, which integrated various aspects, was more comprehensive.

## 3. Materials and Methods 

### 3.1. Samples

The leaves of *R. heterophylla* were collected from Pingwu, where Nü-Er-Cha is frequently plucked and drank, Sichuan Province, China, in May 2015. All the samples were from different geographical origins of Zhennanshan, Yangjiaba, Sizhuya, or Matongmiao, as listed in Table 7. The number of samples was S1–S8, which were obtained by mixing the same weight of six batches of Nü-Er-Cha. The picked leaves were processed by skilled native people so that the moisture of sundried leaves was approximately 12%, and then they were stored in sealed, cool and dry conditions. All the samples were identified by Professor Zhirong Sun (College of Traditional Chinese Medicine, Beijing University of Chinese Medicine).

### 3.2. Chemicals

Acetonitrile (chromatography grade) was acquired from Thermo Fisher Scientific Co., (Shanghai, China) and formic acid was acquired from (highest purity, Beijing Chemical Reagent Co., Beijing, China). The other chemicals were analytical grade.

### 3.3. Sensory Evaluation of Nü-Er-Cha

For sensory assessment, the panel was composed of 50 members from Beijing University of Chinese Medicine (38 female, 12 male, 18–27 years old). Fifty-five individuals participated in the screening session. Firstly, the members were tested for their ability to distinguish the primary taste according to the triangular test, and then the passed individuals were asked to evaluate the strength of taste by means of the ranking test [24]. Finally, 50 assessors were selected and trained to judge the sensory properties of Nü-Er-Cha. Referring to the Methodology of Sensory Evaluation of Tea (GB/T 23776-2009) and The Vocabulary for Sensory Evaluation (GB/T 23776-2018), the present study selected 5 indicators including appearance (rope, evenness, neatness, brightness, leaf color), infusion (infusion color, transparency), infused leaves (texture, openness), taste (sweetness, lubrication) and aroma (clean aroma). The grading system was based on a maximum score of 10, of which the five indicators accounted for 10%, 10%, 30%, 30%, and 20%, respectively. The sensory evaluation of Nü-Er-Cha was carried out at room temperature, and the sample order for each assessor was randomized. The assessors evaluated the appearance of leaves first. Three grams of leaves were infused with 300 mL of freshly boiled tap water for 5 min. Then, the assessors smelled the aroma, assessed infusion color immediately, and drank for the taste evaluation after cooling. Finally, the infused leaves were put into plates for evaluation.

### 3.4. Physicochemical Measurements

#### 3.4.1. Viscosity, Conductivity, pH and Color 

The viscosity tests were carried out by a Pins viscometer (SUNLEX, Shanghai Shenli Glass Instrument Co., Ltd., Shanghai, China), of which the capillary inner diameter is 1.0 mm and the viscosity constant is 0.09730 mm^2^/s^2^. A stopwatch was used to determine the time of infusions flowing between the two scales. The results are presented as mm^2^/s. A conductivity meter (DDS-307A, Yoke Instrument, Shanghai, China) was employed to detect the values of infusions’ conductivity at 25 °C. An electrode was washed by distilled water before measuring each infusion. The conductivity of the tap water was also measured, and the final results are expressed as the differential values (Dcond.) between the conductivity of tap water and infusions. The pH of tea infusions was measured using a portable pH meter (FiveEasy Plus, Mettler Toledo, Shanghai, China). Prior to measuring the pH of infusions, the instrument was calibrated with standard buffer solutions of pH 6.86 and 9.21. The differential values (DpH) of pH between tap water and infusions were also calculated. All determinations were performed in triplicate.

Color was determined based on International Commission on Illumination (CIE) L*a*b* system, where L* for lightness, a* for redness or greenness, and b* for yellowness or blueness. The colorimetric coordinates (L*, a*, b*) were carried out using a colorimeter (CM-5, Konica Minolta, Shanghai, China). The colorimeter was calibrated with a white and a black standard plate before measuring. In the present study, both leaves (grinding to 24 mesh) and infusions were measured and recorded as L*1, a*1, b*1, L*2, a*2 and b*2. The results are reported as the mean of three determinations. 

#### 3.4.2. Establishment of HPLC Fingerprint and HPLC-DAD-ESI-MS/MS Analysis

The reflux extraction was applied in the present work, owing to the lower peak response in HPLC when the tea infusions were used. One-point-five grams of sample powder (passing through a 50-mesh stainless steel sieve) were extracted with 50 mL of water by reflux extraction for 1.5 h to prepare the Nü-Er-Cha solution. The solution was passed through a 0.45 and 0.22 μm porosity, respectively, to conduct HPLC and HPLC-DAD-ESI-MS/MS analyses. 

The Agilent Technologies 1200 Infinity Series equipped with a diode array detector (Agilent Technologies company, Shanghai, China) was employed to establish the fingerprint. The separation was achieved on a Thermo Hypersil Gold C_18_ column (150 mm× 4.6 mm, 5 μm). Mobile phase A was acetonitrile, while mobile phase B was 0.2% formic acid. The gradient elution program of the mobile phases was as follows: 5–10% (A) from 0 to 10 min, 10% (A) for 5 min, 10–20% (A) from 15 to 30 min, 20% (A) for 5 min, 20–30% (A) from 35 to 45 min, 30–45% (A) from 45 to 65 min, and 45–65% (A) from 65 to 70 min. A 10 μL sample solution was injected into autosampler and monitored at 270 nm. The flow rate was set at 1.0 mL/min, and the column temperature was maintained at 25 °C. For the characterization of compounds from Nü-Er-Cha, the HPLC coupled with a quadrupole time-of-flight (Q-TOF) mass analyzer with am electrospray ionization (ESI) ion source (Agilent Q-TOF 6520, Agilent Technologies company, Shanghai, China) was employed. Data were processed with G3335AA MassHunter Qualitative Analysis Software B.04.00. Spectra (Agilent Technologies company, Santa Clara, CA, USA) were recorded in negative ion mode. Applying the same chromatographic conditions with the HPLC fingerprint, the mass spectrometer operating conditions were: Capillary voltage 3500 V, nebulizer pressure 30 psig, dry gas heater temperature 320 °C, fragmentor voltage 80 V, collision energy 30 V, and mass range 100–1000 amu.

### 3.5. TOPSIS Comprehensive Evaluation Method

In the present study, the TOPSIS approach was performed for comprehensively evaluating the quality of Nü-Er-Cha. According to the steps reported by Sun et al. [25], in the first step, a decision matrix X (8 samples × 22 sensory and physicochemical indicators) was established. In the second step, the normalized decision matrix N and the weighted normalized decision matrix W were calculated. In the third step, the positive and negative ideal solutions (the maximum and minimum values of each column, respectively) were determined. In the fourth step, the distance from the positive and negative ideal solution (D^−^, D^+^) were measured by Euclidean distances. In the last step, the relative closeness (R) to the ideal solution was calculated, and the alternatives were ranked based on R values. The R value ranged from 0 to 1. When it was closest to 1, this implied that the option was the closest to the optimal level. Thus, the sample with the highest R value was considered as the best alternative. 

### 3.6. Statistical Analysis

The results of physical parameter measurements were expressed as mean ± standard deviation (*n* = 3). A one-way analysis of variance and Tukey’s tests were performed to identify differences between the means. Differences were considered to be significant at *p* ≤ 0.05. An evaluation of the relationship between sensory evaluation and physicochemical profiles was carried out via a multivariate statistical analysis tools including the Pearson correlation coefficient, principal component analysis (PCA) and partial least squares regression (PLSR). The PCA and PLSR analysis were carried out in the SIMCA software (Umetrics AB, Malmö, Sweden), version 14.1, and the rest were performed using OriginPro 2017 (OriginLab Corporation, Northampton, MA, USA), version b9.4.0.220. 

## 4. Conclusions

Herbal tea consumption shows regional characteristics, and many Chinese herbal teas are underutilized. The quality assessment of herbal teas eventually influences consumer preferences. Sensory evaluation and physicochemical determination are important tools for the quality assessment of tea and herbal teas. Physical parameters, including color, pH and conductivity could characterize sensory traits and have been frequently used to evaluate the quality of tea. In the present study, besides the parameters mentioned above, viscosity was also taken into consideration for quality assessment of Nü-Er-Cha. Clear correlations between total sensory scores and conductivity, viscosity, and pH were observed, indicated that physical parameters could characterize the sensory attributes of Nü-Er-Cha in an objective way. The common peaks at X4 (salicylic acid), X5, X6, X7 (flavonol glycosides) were significantly correlated with the color of Nü-Er-Cha. It is possible to characterize sensory profiles and the interaction among chemical constituents via physical parameters. To comprehensively evaluate the quality of Nü-Er-Cha, the TOPSIS approach applied in comprehensive quality evaluation of traditional Chinese medicine was employed in the present work. The physical parameters, chemical constituents and sensory profiles of Nü-Er-Cha were integrated by the TOPSIS method for comprehensive quality evaluation. The method developed in the present work supplied a reference for the quality assessment of herbal teas. On the one hand, the quality was evaluated based on single aspects including physical parameters, sensory evaluation and chemical constituents. On the other hand, their inner relationship was mined, and the three aspects were integrated by the TOPSIS approach for comprehensive quality assessment. Future research should supplement physicochemical parameters for quality assessment. A larger sample size should also be employed to improve the robustness of method developed.

## Figures and Tables

**Figure 1 molecules-24-03211-f001:**
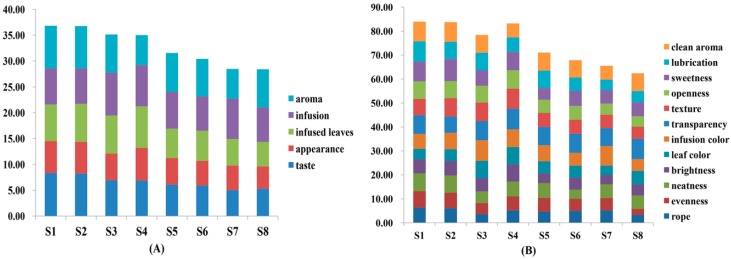
Histogram for the descriptive sensory analysis of Nü-Er-Cha: (**A**) The scores plot of five descriptors; (**B**) the scores plot of twelve indicators.

**Figure 2 molecules-24-03211-f002:**
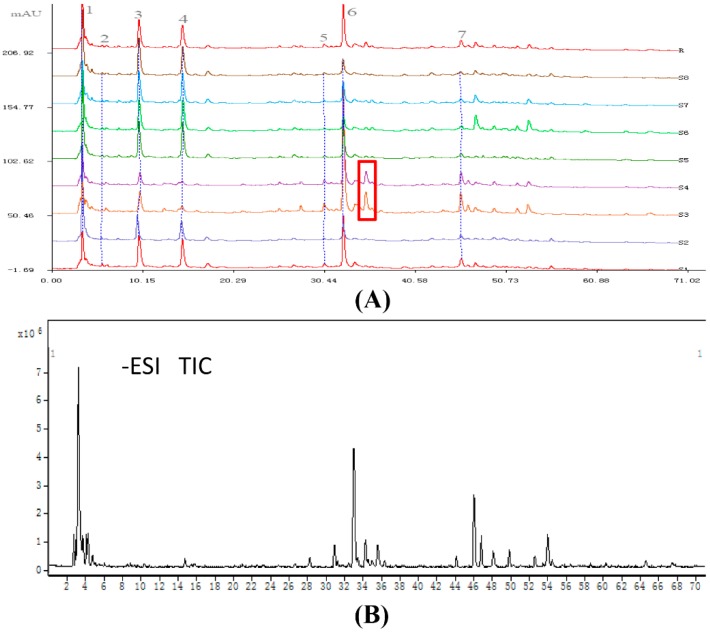
HPLC fingerprint (**A**) and total ion current (TIC) chromatogram (**B**) of Nü-Er-Cha.

**Figure 3 molecules-24-03211-f003:**
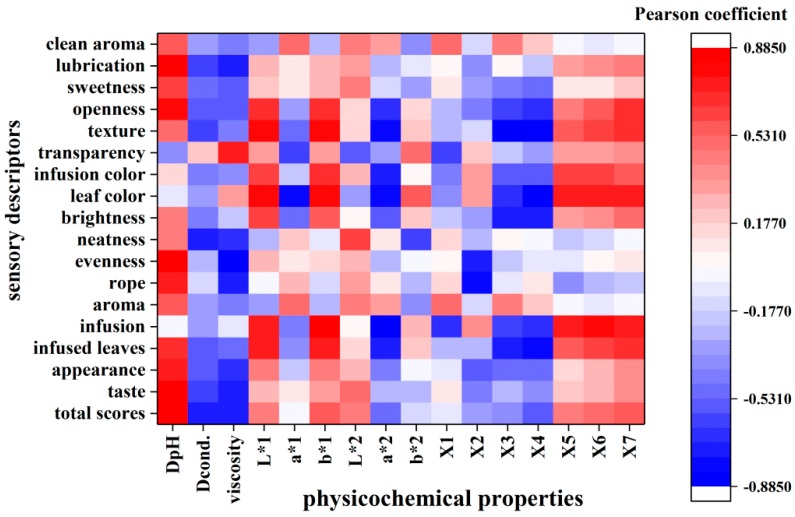
The heat-map based on the Pearson coefficient.

**Figure 4 molecules-24-03211-f004:**
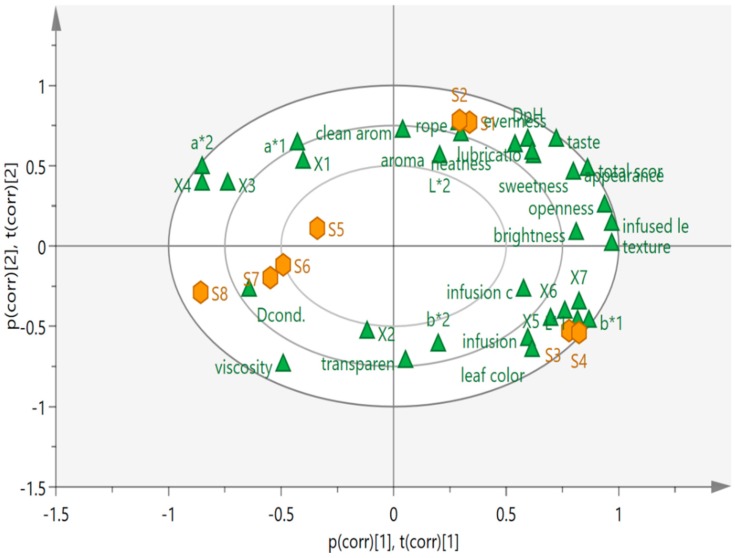
The correlation bi-plot for sensory and physical–chemical parameters.

**Figure 5 molecules-24-03211-f005:**
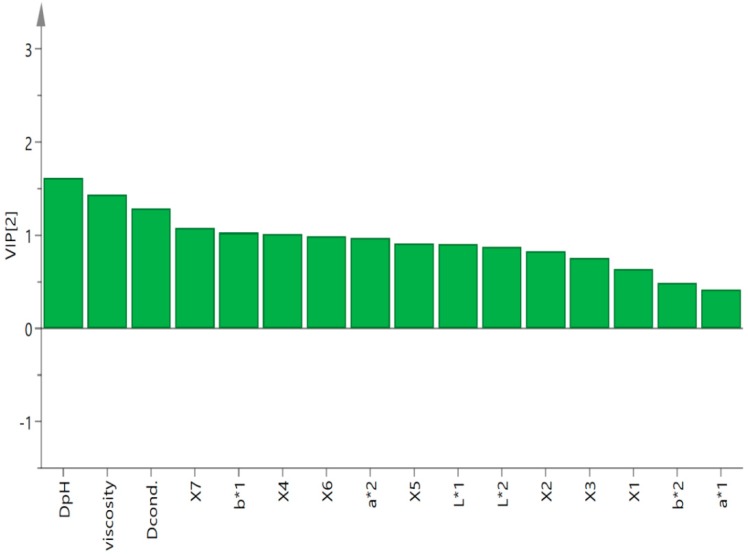
The variable importance to projection (VIP) of the partial least squares regression (PLSR) analysis of total sensory evaluation scores and physical–chemical parameters.

**Table 1 molecules-24-03211-t001:** The physicochemical properties analysis of Nü-Er-Cha ^A^.

Samples	DpH ^B^	Dcond. ^C^ (μs/cm)	Viscosity (m^2^/s)				Color		
				Leaves ^D^			Infusions ^E^		
				L*1	a*1	b*1	L*2	a*2	b*2
S1	0.74 ± 0.05 ^a^	47 ± 12 ^cd^	1.53 ± 0.19 ^b^	33.29 ± 0.01 ^ab^	2.78 ± 0.02 ^a^	19.02 ± 0.01 ^c^	98.32 ± 0.00 ^a^	0.59 ± 0.00 ^abc^	6.08 ± 0.00 ^c^
S2	0.65 ± 0.04 ^b^	40 ± 7 ^d^	1.53 ± 0.05 ^b^	34.05 ± 0.01 ^ab^	2.10 ± 0.02 ^c^	18.52 ± 0.04 ^d^	98.08 ± 0.00 ^ab^	0.69 ± 0.01 ^abc^	6.28 ± 0.01 ^bc^
S3	0.61 ± 0.02 ^bc^	41 ± 6 ^cd^	1.56 ± 0.09 ^ab^	43.28 ± 0.02 ^ab^	1.22 ± 0.00 ^f^	26.76 ± 0.05 ^a^	97.51 ± 0.00 ^abc^	−2.69 ± 0.01 ^bc^	13.19 ± 0.02 ^abc^
S4	0.57 ± 0.02 ^c^	56 ± 8 ^cd^	1.57 ± 0.04 ^ab^	43.41 ± 0.00 ^a^	−0.58 ± 0.01 ^h^	24.82 ± 0.02 ^b^	95.73 ± 0.01 ^bc^	−2.54 ± 0.01 ^bc^	18.35 ± 0.04 ^a^
S5	0.59 ± 0.02 ^bc^	61 ± 7 ^bc^	1.55 ± 0.06 ^ab^	33.34 ± 0.01 ^ab^	1.09 ± 0.02 ^g^	17.59 ± 0.02 ^f^	96.07 ± 0.00 ^abc^	0.32 ± 0.00 ^abc^	14.95 ± 0.02 ^abc^
S6	0.60 ± 0.01 ^bc^	113 ± 6 ^a^	1.58 ± 0.33 ^ab^	36.35 ± 0.01 ^ab^	1.59 ± 0.01 ^e^	17.61 ± 0.03 ^f^	95.33 ± 0.00 ^c^	0.88 ± 0.01 ^ab^	17.58 ± 0.02 ^ab^
S7	0.50 ± 0.01 ^c^	77 ± 5 ^b^	1.56 ± 0.05 ^ab^	32.55 ± 0.01 ^ab^	2.44 ± 0.02 ^b^	18.08 ± 0.03 ^e^	97.60 ± 0.00 ^abc^	0.72 ± 0.00 ^abc^	7.49 ± 0.00 ^abc^
S8	0.43 ± 0.02 ^d^	66 ± 3 ^bc^	1.60 ± 0.03 ^a^	28.68 ± 0.42 ^b^	1.78 ± 0.02 ^d^	16.04 ± 0.03 ^g^	96.88 ± 0.00 ^abc^	2.03 ± 0.00 ^a^	8.18 ± 0.01 ^abc^

^a–h^ One-way ANOVA conducted at α = 0.05. Post-hoc test conducted using Tukey’s honestly significant difference test. ^A^ Values are expressed in terms of mean ± standard deviation. N = 3 panelists. ^B^ The differential values between the conductivity of tap water and infusions. ^C^ The differential values between the conductivity of tap water and infusions. ^D^ The color parameters of leaves. ^E^ The color parameters of infusions.

**Table 2 molecules-24-03211-t002:** The relative peak area of common peaks.

Common Peaks	S1	S2	S3	S4	S5	S6	S7	S8
X1	0.5926	5.0868	0.2053	0.3683	2.1660	3.9654	1.5900	3.0771
X2	0.0290	0.1588	0.0348	0.0314	0.0812	0.1189	0.1600	0.2555
X3	0.7450	1.5302	0.1738	0.1919	1.3710	2.3854	1.4628	2.2161
X4	0.6424	1.4100	0.0536	0.0662	1.5070	2.7684	1.6495	2.0251
X5	0.0650	0.0987	0.0736	0.0651	0.0627	0.1103	0.0867	0.1200
X6	1.0000	1.0000	1.0000	1.0000	1.0000	1.0000	1.0000	1.0000
X7	0.2055	0.2393	0.1535	0.1906	0.1574	0.2434	0.1593	0.2050

**Table 3 molecules-24-03211-t003:** Results of similarity analyzes of the HPLC fingerprint of Nü-Er-Cha.

	S1	S2	S3	S4	S5	S6	S7	S8	R
S1	1	0.696	0.801	0.829	0.893	0.810	0.910	0.849	0.888
S2	0.696	1	0.400	0.514	0.910	0.879	0.806	0.906	0.915
S3	0.801	0.400	1	0.988	0.536	0.427	0.561	0.467	0.548
S4	0.829	0.514	0.988	1	0.618	0.518	0.627	0.556	0.634
S5	0.893	0.910	0.536	0.618	1	0.950	0.953	0.975	0.989
S6	0.810	0.879	0.427	0.518	0.950	1	0.948	0.953	0.963
S7	0.910	0.806	0.561	0.627	0.953	0.948	1	0.964	0.971
S8	0.849	0.906	0.467	0.556	0.975	0.953	0.964	1	0.987
R	0.888	0.915	0.548	0.634	0.989	0.963	0.971	0.987	1

**Table 4 molecules-24-03211-t004:** The compounds identified by HPLC-DAD-ESI-MS/MS.

Compounds	Molecular Formula	Observed Mass (*m*/*z*)	Expected Mass (*m*/*z*)	Mass Error (ppm)	RT (min)	[M − H]^−^ (*m*/*z*)	MS/MS Fragment Ions (*m*/*z*)
**flavonoids**							
3′,5,7-Trihydroxy-4′-methoxyflavone 7-*O*-rutinoside	C_28_H_32_O_15_	608.1738	608.1741	−0.49	46.779	607.1699	461.6880, 299.0577, 152.016, 148.6852
quercetin	C_15_H_10_O_7_	302.0429	302.0427	0.91	48.095	301.0316	152.0082, 153.0085, 121.0286
quercetin 3-methyl ether	C_16_H_12_O_7_	316.0588	316.0583	1.3	54.461	315.0529	300.0259, 272.0315, 152.0128
rhamnezin	C_17_H_14_O_7_	330.0738	330.074	−0.41	67.703	329.0734	314.0509, 299.0226, 271.0245, 227.0336
kaempferol	C_15_H_10_O_6_	286.0484	286.0477	1.62	54.037	285.0414	257.0445, 229.0508, 121.026, 93.0356, 65.00333
rhamnocitrin	C_16_H_12_O_6_	300.0636	300.0634	1.01	67.449	299.0624	271.0607, 243.0695, 166.0197
quercetin galactoside(3-*O*;7-*O*)	C_21_H_20_O_12_	464.0953	464.0955	0.2	32.434	463.088	301.0325, 273.0334, 245.0515, 136.9104
kaempferol 3-*O*-glucoside/galactoside	C_21_H_20_O_11_	448.1011	448.1006	0.51	37.484	285.0333	257.0445, 229.0511, 152.0099
quercetin diglycoside with a rhamnose and a galactose/glucose)	C_27_H_30_O_16_	610.1532	610.1534	−0.36	31.288	609.1523	301.0313, 273.0802, 152.8107, 136.9585
quercetin triglycoside with a galactose/glucose and two rhamnoses (X5)	C_33_H_40_O_20_	756.2144	756.2113	0.88	30.906	755.2144	301.0349, 463.0765, 152.4898, 137.3381
quercetin 3-methyl ether 7-*O*-galactopyranoside	C_22_H_22_O_12_	478.1106	478.1111	−0.34	51.788	477.1018	315.0430, 300.0189, 165.0141, 137.5443, 109.0301
kaempferol triglycoside with a galactose/glucose and two rhamnoses (X6)	C_33_H_40_O_19_	740.2158	740.2164	0.07	32.901	739.2122	285.0403, 57.0435, 229.0946, 152.0061, 120.0288
rhamnocitrin diglycoside with a rhamnose and a galactose/glucose)	C_28_H_32_O_15_	608.1738	608.1741	−0.83	47.012	607.1597	299.0577, 255.7552, 165.5555, 119.9104
rhamnocitrin triglycoside with a galactose/glucose and two rhamnoses (X7)	C_34_H_42_O_19_	754.2335	754.232	1.01	40.015	753.2289	299.0507, 271.0618, 255.0335, 165.0185
rhamnazin triglycoside with a galactose/glucose and two rhamnoses	C_35_H_44_O_20_	784.2431	784.2426	0.71	46.822	783.236	329.0661, 314.0414, 286.0448,
rhamnazin diglycoside with a rhamnose and a galactose/glucose)	C_29_H_34_O_16_	638.1852	638.1847	0.61	47.586	637.1799	329.0737, 314.0414, 299.0407
rhamnetin triglycoside with a galactose/glucose and two rhamnoses	C_34_H_42_O_20_	770.2276	770.2269	0.48	34.280	769.2203	315.0507, 300.0245, 272.0272, 165.5149, 137.721
**anthroquainones**							
emodin	C_15_H_10_O_5_	270.053	270.0528	0.49	64.562	269.0462	225.0574, 241.0480,
emodin-3-*O*-rhamnoside	C_21_H_20_O_9_	416.1109	416.1107	0.4	64.605	415.1148	269.0459, 241.0525, 197.0590
emodin glucoside(1-*O*;8-*O*)	C_21_H_20_O_10_	432.1065	432.1056	2.01	49.708	431.094	269.0461, 240.0423, 241.0514, 225.0562
emodin 3-*O*-(di-*O*-acety) rhamnoside	C_25_H_24_O_11_	500.1321	500.1319	0.48	58.069	499.1353	269.0411, 240.0461, 212.0476
**organic acids**							
malic acid (X1)	C_4_H_6_O_5_	134.0216	134.0215	0.76	3.701	133.0128	115.0039, 71.047,
gallic acid (X2)	C_7_H_6_O5	170.0213	170.0215	−1.48	5.992	169.0127	152.0945, 125.0271
protocatechuic acid (X3)	C_7_H_6_O_4_	154.0264	154.0266	−1.74	10.391	153.0181	109.0283, 91.0185, 63.0259
salicylic acid (X4)	C_7_H_6_O_3_	138.0318	138.0317	−1.02	15.435	137.0245	93.0348, 65.0386,51.0226

**Table 5 molecules-24-03211-t005:** The rankings according to different factors used in the technique for order preference by a similarity to ideal (TOPSIS) approach.

Rankings	Sensory and Physicochemical Parameters	Physical Parameters	Chemical Constituents	Sensory Profiles
1	S3	S6	S3	S2
2	S4	S4	S4	S1
3	S1	S3	S1	S3
4	S5	S5	S5	S4
5	S6	S7	S8	S5
6	S8	S1	S7	S6
7	S7	S8	S6	S8
8	S2	S2	S2	S7

**Table 6 molecules-24-03211-t006:** Final rankings by the TOPSIS approach.

Rankings	Samples	D+	D−	R
1	S3	0.28362	0.57927	0.67131
2	S4	0.42034	0.33637	0.44451
3	S1	0.47427	0.27501	0.36703
4	S5	0.53332	0.26859	0.33493
5	S6	0.57201	0.27990	0.32855
6	S8	0.57241	0.23610	0.29202
7	S7	0.56961	0.21190	0.27114
8	S2	0.59041	0.21846	0.27008

**Table 7 molecules-24-03211-t007:** The sources of Nü-Er-Cha.

No.	Location	Batches Number	Date
S1	Zhennanshan	6	2015.05
S2	Zhennanshan	6	2015.05
S3	yangjiaba	6	2015.05
S4	yangjiaba	6	2015.05
S5	Sizhuya	6	2015.05
S6	Sizhuya	6	2015.05
S7	Matongmiao	6	2015.05
S8	Matongmiao	6	2015.05

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
