# Peer review of "Physicochemical Aspects and Sensory Profiles as Various Potential Factors for Comprehensive Quality Assessment of Nü-Er-Cha Produced from Rhamnus heterophylla Oliv."

_molecules, 2019, doi:10.3390/molecules24183211_

Round 1
Reviewer 1 Report
Congratulations; a very god work. Here are a few f my comments and suggestions:
Section 3 (Materials and Methods) should come before Results and Discussion. It is always very helpful to the reader to see first the methods and equipment used in the work to clearly understand the results. Table 1 showing the results of the physicochemical properties of Nu-Er-Cha is not clear; you must separate the color results of leaves and infusion, and indicate what a*1, b*1, and a*2 and b*2 indicate. You must also define what DpH and Dcond. The Spider plot presented on Figure 1 does not seem to be adequate to demonstrate the differences of sensory analysis results. Use different type of plot to demonstrate the differences among the twelve indicators. You indicate that S3 and S4 showed significant similarities based on the correlation coefficient analysis of the HPLC results, indicating that both samples contained rhamnetin triglucoside (galactose/glucose and rhamnose); but you do not show the different compounds (flavonoids, anthraquinones and/or phenolic compounds) present in each of the tea samples. It is important to know the chemical constituents of each type of sample. You have demonstrated by TOPIS approach that physical characteristics and sensory traits are important for the evaluation of the quality of the tea samples, but you must also show the differences in the concentration of each of the chemical constituents in the tea samples in order to judge the medical value of the each type of teaAuthor Response
Thank you very much for your valuable comments on our manuscript. Please see the attachment.

Reviewer 2 Report
Dear authors,
this is an interesting manuscript; please address the following comments:
please have a native English speaker review the manuscript for language Sensory methods: why was it necessary to have 50 trained panelists? That seems excessive. Please provide definitions and reference materials for the attributes evaluated. How were the panelists calibrated for the attributes and samples evaluation? In the stat analysis section you mention that you ran all samples in triplicate and have provided appropriate stat analysis, however there is no indication of standard deviations or ANOVA analysis for the sensory portion. It is not clear why you have run PCA on all of the data collected. Please provide justification. Why did you run both PCA and PLS analysis? Would just one of these not suffice? Provide details of statistical analysis in section 3.6.Author Response
Thank you very much for your valuable comments on our manuscript. Please see the attachment.
